



# Distinct Photochemistry in Glycine Particles Mixed with Different Atmospheric Nitrate Salts

Zhancong Liang[1,2], Zhihao Cheng[1], Ruifeng Zhang[1,2], Chak K. Chan[1, 3*]

[1]School of Energy and Environment, City University of Hong Kong, Hong Kong, China
[2]City University of Hong Kong Shenzhen Research Institute, Shenzhen, China
[3]Division of Physical Science and Engineering, King Abdullah University of Science and Technology, Thuwal, 23955-6900, Saudi Arabia

*Correspondence to*: Chak K. Chan (chak.k.chan@cityu.edu.hk; chak.chan@kaust.edu.sa)

**Abstract.** Particulate Free amino acids (FAAs) are essential components of organonitrogen that have critical climate impacts, but they are usually considered stable end products from protein degradation. In this work, we investigated the decay glycine (GC) as a model FAA under photolysis of different particulate nitrate salts using an in-situ micro-Raman system. Upon changes of the relative humidity (RH) cycle between 3 and 80% RH, ammonium nitrate (AN)+GC mixed particles did not exhibit any phase change, whereas sodium nitrate (SN)+GC mixed particles crystallized at 60% and deliquesced at 82% RH. Under light illumination at 80% RH, AN+GC particles showed almost no spectral changes, while rapid decays of glycine and nitrate were observed in SN+GC particles. The interactions between nitrate and glycine in AN+GC particles suppressed crystallization, but also hindered nitrate photolysis and glycine decay. On the other hand, glycine formed a complex with Na$^+$ in deliquescent SN+GC particles and allowed unbonded nitrate to undergo photolysis and trigger glycine decay, though nitrate photolysis was greatly hindered upon particle crystallization. Our work provides insights into how FAAs may interact with different nitrate salts under irradiation and lead to distinct decay rates, which facilitates further investigations on their atmospheric lifetime estimation.

## 1 Introduction

Free amino acids (FAAs) are essential components of atmospheric particles with wide sources, including direct bio-emission, degradation of proteinaceous materials, and biomass burning (Ren et al., 2018; Matos et al., 2016; Zhu et al., 2021; Li et al., 2022a; Liu et al., 2017). The concentrations of FAAs in terrestrial and marine near-surface atmospheres generally range from a few to several hundred ng m$^{-3}$ (Helin et al., 2017; Matos et al., 2016). FAAs play important roles in the climate-related properties of atmospheric particles, such as hygroscopicity and cloud condensation nuclei (CCN) activity (Chan et al., 2005; Kristensson et al., 2010; Marsh et al., 2017), and atmospheric nitrogen cycling (Mopper et al., 1987). Besides, FAAs support the biological activity in the aerosol particles (Helin et al., 2017) and impact human health after inhalation (Hu et al., 2020). While the importance of atmospheric FAAs was well-recognized, previous atmospheric studies mainly focused on the regional and seasonal variations of the abundance of particulate FAAs (Ren et al., 2018; Helin et al., 2017; Song et al., 2017). FAAs were suggested as the stable end products of the degradation of protein (Li et al., 2022a). However, a few studies have reported



the reactivity of atmospheric FAAs with dicarbonyls, to form light-absorbing compounds via oligomerization under dark (De Haan et al., 2011; Haan et al., 2009). The photochemical evolution of FAAs remains less explored.

Nitrate is ubiquitous in atmospheric particles (Chan et al., 2008), and it can generate various oxidants, including OH radicals, via photolysis (Gen et al., 2022; Scharko et al., 2014; Benedict et al., 2017). Oxidants from particulate nitrate photolysis could

oxidize gaseous precursors to secondary inorganic and organic aerosols (Gen et al., 2022; Gen et al., 2019; Zhang et al., 2021; Zhang et al., 2022) and change the morphology of the particles (Liang et al., 2021). Correlation analysis of field measurements suggests that particulate FAAs and nitrate could be from the same sources (Xu et al., 2019; Gao et al., 2021), and there were also laboratory studies on the hygroscopicity of FAAs-nitrate mixed particles (Ashraf et al., 2021; Wang et al., 2018). While external oxidants such as gaseous ozone and OH radicals need to diffuse across the interfacial layers, nitrate photolysis

generates oxidants inside the particles to oxidize the particulate components (Liang et al., 2021). These characteristics make effective FAAs oxidation triggered by nitrate photolysis possible. However, it is unclear how FAAs would chemically evolve under nitrate photolysis, though it is well-known that OH radicals can react with FAAs in water treatment research (Berger et al., 1999; Acero et al., 2000). Recently, Wen et al. (2022) demonstrated that aqueous OH oxidation could be a significant sink of atmospheric FAAs (Wen et al., 2022). Nevertheless, while these results were helpful to dilute aqueous systems, the reactions

in the particle phase could be different due to the elevated concentrations that facilitate molecular interactions. Specifically, glycine (GC), which is the most abundant FAA in atmospheric particles (mole ratio of GC to total FAAs = 0.17-0.49) (Li et al., 2022a; Ren et al., 2018; Song et al., 2017; Zhang et al., 2003; Zhu et al., 2020), could bind with atmospheric-relevant inorganics such as sulfate and nitrate (Ashraf et al., 2021). The phase state of the particles can also play an essential role in nitrate photolysis (Liang et al., 2021; Gen et al., 2022).


In this paper, we first performed relative humidity (RH)-cycle to characterize the phase transition behavior of mixed particles of ammonium nitrate (AN) and glycine as well as sodium nitrate (SN) and glycine. behaviors. Then, we examined the photochemistry of glycine triggered by the photolysis of AN and SN at different RHs. The phase transition behaviors of AN+GC and SN+GC mixed particles and the kinetics of glycine decay under UV illumination are significantly different.

## 2 Experimental

### 2.1 Photochemical aging of the droplets

Mixed solutions (1% wt) of AN (>99%, Sigma-Aldrich) or SN (>99%, Arcos) and glycine (Biological analysis level, Chemcruz™) were prepared in ultrapure water (Milli-Q). We used a mole ratio of 1:1 for glycine and nitrate in all our experiments. Mixed solutions were atomized using a piezoelectric particle generator (model 201, Uni-Photon Inc.) coupled

with a quartz tip (Micro-Fab, orifice diameter = 80 μm). Droplets were deposited on a hydrophobic fluorocarbon substrate (model 5793, YSI Inc.) and placed in an aerosol flow cell (Liang et al., 2021; Liang et al., 2022b). The flow cell has two



windows for in situ Raman analysis (top) and ultraviolet (UV) illumination (bottom). The schematic of the experimental setup is shown in the Supporting Information (Figure S1). RH inside the flow cell was controlled by mixed dry and wet synthetic air (Linde) and monitored by a digital RH sensor (HC2-C05, ROTRONIC AG, Switzerland). The deposited droplets were

photolyzed for 8 h using a 300 nm light-emitting diode (LED) lamp (M300L4, Thorlabs), through the bottom window of the flow cell. The photon flux received by deposited particles in the flow cell was determined to be $1.2 \times 10^{15}$ photons cm$^{-2}$ s$^{-1}$ by 2-nitrobenzaldehyde (2NB, >99.0%, Acros Organics). A detailed description can be found in our previous work (Liang et al., 2021). The effective incident light flux (photons cm$^{-2}$ s$^{-1}$) used in our study was comparable to that received by nitrate in the atmosphere on a typical clean day.


**2.2 In-situ Raman and microscopic characterization**

A Raman spectrometer (EnSpectr R532, EnSpectr) with a 20−30 mW 532 nm laser and holographic diffraction grating with 1800 grooves/mm was used to characterize the particle *in-situ* during phase transition behavior measurement and photoreactions of the particles. The Raman spectrometer was coupled with an optical microscope (CX41, Olympus) to acquire

Raman spectra at 100−4000 cm$^{-1}$ at a resolution of 4 cm$^{-1}$. A 50× objective lens with a numerical aperture of 0.35 (SLMPLN50X, Olympus) was used to guide the laser onto the sample. For the phase transition measurements, we evaporated and then humidified the droplets by decreasing and increasing the RH gradually. Images of the particles were captured, and the Raman spectra were recorded after equilibrium was reached. The size of the equilibrated droplets at 80% RH were 41±15 μm. The in-situ Raman analysis focused on single particles of ~40 μm, while approximately 1300 particles were used for off-

line analysis (will be discussed later). Particle composition during photoreactions was monitored using Raman measurement every hour for 8 hours. The integration time for each spectrum is 5 s. Glycine can form light-absorbing mesocluster in droplets and trigger photosensitization to degrade themselves at 532 nm, the Raman excitation wavelength in our experiments (Ishizuka et al., 2023). This mechanism plays a minor role in our system as the glycine decay without the LED was negligible.

**2.3 Off-line chemical analysis of the particle extract**

Particle-loaded substrates were extracted using 1 mL MilliQ water after photoreactions. The water extract was analyzed by ion chromatography (IC). The IC protocol was the same as our previous work (Liang et al., 2022c). After equilibrating at 80% RH, the initial particle pH was measured by a pH indicator combined with RGB-based colorimetric analyses using a model of $G−B$ ($G$ minus $B$) vs. pH$^2$, according to Craig et al. (Craig et al., 2018). The particles were considered at equilibrium when

the size became unchanged (±2%) for 30 min. The nitrate and glycine concentrations were obtained according to Eq. 1, where "a" is the scaling factor determined by calibration, and A refers to the integrated area of the corresponding peak using Gaussian fitting (Igor Pro 8). The wavenumber ranges used for integration were 850-950 cm$^{-1}$ for glycine and 980-1100 cm$^{-1}$ for nitrate, respectively.

$$[X]_{particle} = a \times (A(X) / A(OH)_{3400cm^{-1}}) \qquad \text{X is nitrate, glycine} \qquad (1)$$






## 2.4 Estimation of nitrate photolysis rate constant and percentage glycine decay

The maximum RH reached in the flow cell was 96%, which yields a solute concentration in particles higher than 1 M. Thus, data of diluted systems (0.01, 0.1, and 0.5 M) were obtained from kinetic measurements of aqueous solutions. The schematic of the custom-made aqueous reactor is shown in Figure S2. Synthetic air was introduced to the aqueous reactor at 0.1 L/min.

AN+GC and SN+GC solution (0.01, 0.1, or 0.5 M) were added to the aqueous reactor and illuminated by 300 nm LED through a quartz window on the top of the reactor. The photon flux received by the solution was determined to be $0.7 \times 10^{15}$ photons cm$^{-2}$ s$^{-1}$ by 2-NB, ~60% of that flux found for deposited particles in the flow cell. Therefore, we sampled the aliquots from the aqueous solution after 13.3 h of irradiation. After sampling, the glycine concentration was immediately determined using the pre-column derivatization HPLC method described by Matsumoto et al. (Matsumoto et al., 2021). The nitrate concentration

was determined by IC.

The apparent nitrate photolysis rate constant J (s$^{-1}$) was estimated as Eq. 2:

$$\frac{d[NO_3^-]}{dt} = -J \times [NO_3^-] \tag{2}$$

This is a low estimate of J since glycine oxidation likely generates secondary nitrate (Berger et al., 1999). We estimated the percentage GC decay to indicate the effectiveness of the decay under different conditions based on [GC] measured before and

after irradiation (after 8 h for deposited particles and 13.3 h for solutions). The percentage GC decay in crystalline SN+GC particles was estimated directly by the GC peak as the water peak was not available.

## 2.5 Estimation of the water to glycine mole ratio

Though still under debate, the water-to-glycine mole ratio was reported to play a crucial role in determining the form of neutral

glycine (Aikens et al., 2006; Tortonda et al., 1996). Specifically, several water molecules were required to stabilize a neutral glycine molecule as a zwitterion, with the amino group protonated and the carboxylic group deprotonated. Otherwise, the neutral glycine would exist as non-ionized molecules. Herein, we estimate the water-to-glycine mole ratio in AN+GC and SN+GC particles and discuss the potential form of glycine, which may play a role in photochemistry.

$$GF = \frac{V_{wet}}{V_{dry}} = \frac{(m_w + m_{dry})/\rho_{wet}}{\frac{m_{dry}}{\rho_{dry}}} = \left(1 + \frac{m_w}{m_{dry}}\right) \times \frac{\rho_{dry}}{\rho_{wet}} = (1 + \frac{M_w}{M_{dry}} \times \frac{n_w}{n_{dry}}) \times \frac{\rho_{dry}}{\rho_{wet}} \tag{4}$$

$$GF = \frac{V_{wet}}{V_{dry}} = \frac{(d_{wet})^3}{(d_{dry})^3} \tag{5}$$

GF is the volumetric growth factor (i.e., the volume ratio of a wet droplet to dry particles at specific RH), and V, m, ρ, n, M, and d represent the volume, mass, density, mole number, molar mass and diameter of the particles, respectively. The subscripts dry, wet, and w denote dry particles (i.e., solutes), wet particles, and water, respectively. Note that the estimation of GF using equation 5 assumes spheric particles on the hydrophobic substrate, d$_{dry}$ was estimated by averaging two measured diameters from orthogonal directions. For an initially non-spherical particle to form a droplet upon RH increase, the estimation of GF by

equation 5 would be a slight overestimation (Matsumura et al., 2007). Then, the water-to-glycine mole ratio (WGR) can be solved by:



$$WGR = \frac{1}{2} \times \frac{n_w}{n_{dry}} = \left[\frac{(d_{wet})^3}{(d_{dry})^3} \times \frac{\rho_{wet}}{\rho_{dry}} - 1\right] \times \frac{M_{dry}}{M_w} \tag{6}$$

The mean molar mass of glycine and the nitrate salts was used as $M_{dry}$. $\rho_{dry}$ of SN+GC particles was available in the literature

(Suresh et al., 2010), while $\rho_{dry}$ of AN+GC particles and $\rho_{wet}$ of both particles were estimated based on the simple volume

additivity rule (Equation 9) (Ha et al., 1999; Tang, 1997). $\rho_{GC}$, $\rho_{AN,}$ and $\rho_{SN}$ were the densities of pure aqueous solutions at the

total solute mass fraction X of the mixed solution obtained from the literature (Venkatesu et al., 2007) and E-AIM prediction

(Clegg et al., 1998).

$$\frac{1}{\rho_{wet}} = \frac{X_{GC}}{\rho_{GC}} + \frac{X_{AN}}{\rho_{AN}} / \frac{X_{SN}}{\rho_{SN}} \tag{7}$$

**3 Results and discussions**

**3.1 Phase transition behavior of nitrate-glycine mixed particles**

We first examined the phase transition behavior of the nitrate-glycine mixed particles without UV illumination. Figure 1 shows

the images and Raman spectra of AN+GC and SN+GC particles undergoing an evaporation-humidification cycle. The black

line in Figure 1b shows the Raman spectra of an AN+GC mixed droplet at 85% RH. The $\nu(NO_3^-)$ peaks are at ~730 cm$^{-1}$ and

~1040 cm$^{-1}$ (Ling et al., 2007). The C-C stretching and C-N stretching peaks are located at ~890 cm$^{-1}$ (Socrates, 2004). In

1300-1450 cm$^{-1}$, there are overlapping peaks from C-H vibration in different chemical environments. Peaks at 2970 cm$^{-1}$ and

3020 cm$^{-1}$ show the antisymmetric and symmetric stretching of $CH_2$, respectively. The two broad peaks at 3250 cm$^{-1}$ and 3450

cm$^{-1}$ are from the stretching of OH, indicating the presence of liquid water (Furić et al., 1992). The $\nu(NH_4^+)$ also contributed

to the peak at 3250 cm$^{-1}$.


Upon RH decrease from 85% to 3%, the AN+GC particle shrank but remained spherical, suggesting the particle gradually lost

water and became dry amorphous solids (Figure 1a). Crystallization did not occur since there was no sudden decrease in the

full width at half maxima (FWHM) of the nitrate and glycine peaks (Liang et al., 2021; Surovtsev et al., 2012; Liang et al.,

2022a). Besides, the 3450 cm$^{-1}$ peak diminished, suggesting that the particle lost water without phase transition, consistent

with the literature (Wang et al., 2022). Although some studies reported an absence of efflorescence RH (ERH) in pure AN

(Zuend et al., 2011; Lightstone et al., 2000), the ERH of pure GC was 61.7% (Wang et al., 2022). Adding crystallizable

organics such as succinic acid to AN would also promote its crystallization (Lightstone et al., 2000). The absence of a phase

transition in the AN+GC particles can be attributed to the chaotropic nature of AN, which results in the "salting-in" effect of

glycine and the gradual but not abrupt evaporation of water (Ashraf et al., 2021), without crystallization. As an amino acid,

GC has a proton-donating carboxyl (COOH) group and a proton-accepting amino ($NH_2$) group. The latter can form hydrogen

bonding with nitrate to suppress crystallization (Wang et al., 2022). This was supported by the FWHM increases of GC and

nitrate peaks as RH decreased (Figure S3) due to intensified molecular interaction. The particles are likely dehydrated at 3%





RH due to the absence of OH peaks from 3200 to 3500 cm$^{-1}$. As RH increased, the particle took up water again and grew. There was no spectral change other than the increase of the O-H peak at 3450 cm$^{-1}$ (Guo et al., 2010).


For the SN+GC particle, the particle size decreased as RH decreased from 84% to 60% (Figure 1c). The black line in Figure 1d shows the Raman spectrum of an SN+GC mixed droplet at 84% RH, which is almost identical to that of an AN+GC mixed droplet at 85% RH. Interestingly, different from AN+GC particles, a phase transition from liquid to crystalline solid was observed at 54% RH. The Raman spectra show a redshift of the nitrate peak at 730 cm$^{-1}$ to 710 cm$^{-1}$ and a blue shift of the

1046 cm$^{-1}$ peak to 1051 cm$^{-1}$, attributable to the formation of glycine-sodium nitrate crystal (GSN) (Figure S4)(Gujarati et al., 2015). Two new peaks are attributed to the NH$_3$ rocking mode at 1100 cm$^{-1}$ after crystallization, likely due to the more restricted vibration in the crystalline lattice than in the aqueous droplet (Jentzsch et al., 2013). The FWHM of the CH$_2$ peaks at 1300 cm$^{-1}$ and 3020 cm$^{-1}$ also decreased after crystallization. As RH further decreased to 3%, no noticeable change in appearance was observed. The SN+GC crystal returned to a droplet at 82% RH after humidification.


### 3.2 Photochemistry of glycine with different nitrate salts

The different phase transition behaviors observed in AN+GC and SN+GC particles reflect the role of molecular interaction in determining the physicochemical properties of the particles. To examine if such interactions could play a role in the chemical reactivity of the particles, we exposed the AN+GC and SN+GC particles to UV irradiation at 80% RH.


The Raman spectral characteristics of AN+GC particles only show slight changes from 0 to 8 h irradiation (Figure 2a). Offline IC analysis also shows no new product formed (Figure S5). However, the spectra of SN+GC particles show apparent changes upon light irradiation. Overall, glycine peaks, including C-N/C-C (890 cm$^{-1}$) and CH$_2$ (1325 cm$^{-1}$, 1425 cm$^{-1}$, 2970 cm$^{-1}$ and 3020 cm$^{-1}$) (Kumar et al., 2005), decreased but peaks at 920 cm$^{-1}$ (C-C) and 1350 cm$^{-1}$ (C-O) attributable to acetate and formate,

respectively, emerged (Figure S5, S6) (Zhang et al., 2021). The rising peaks at 1350 and 2925 cm$^{-1}$ correspond to amide and ammonia or/and amine (Philipsen et al., 2013; Socrates, 2004). Besides, nitrite was also found in the particle extracts using IC. Nitrate photolysis directly generates HNO$_2$/NO$_2^-$ (Gen et al., 2022), and the reaction between the two other nitrate photolysis products, NO$_2$ and OH$^-$, would also form nitrite (Pei et al., 2022).

### 3.3 Nitrate photochemistry of AN+GC and SN+GC particles

The efficiencies of nitrate photolysis in the two mixed systems were different. The fitted apparent nitrate photolysis rate constant of SN+GC particles at 80% RH was $9 \times 10^{-6}$ s$^{-1}$ (R$^2$ = 0.95), 4.5-folds higher than that of AN+GC particles (Figure S7). The faster nitrate photolysis in SN+GC particles likely contributed to the faster glycine decay, and the molecular interactions may explain the discrepancy in nitrate photolysis rate constant between SN+GC and AN+GC particles.






Amino acid nitrate can form through (water-mediated) hydrogen bonding between nitrate from AN and the protonated amino group of glycine (Figure 3a) (Wang et al., 2022; Ashraf et al., 2021). As a result, the amino acids and nitrate ions are bounded in an extensive three-dimensional hydrogen-bonded lattice as supported by XRD analysis (Hudson et al., 2009), in which nitrate photolysis could be hindered (Vimalan et al., 2010). We envision that such interactions exist in the AN+GC system.

On the other hand, the $COO^-$ of glycine can bind with SN via $Na^+$ directly to form a bidentate complex (Figure 3a) (Moision et al., 2002; Aziz et al., 2008; Selvarani et al., 2022), leaving nitrate unbonded. The nitrate peak in SN+GC particles split into two (Figure 3b). One has the same Raman shift as nitrate in AN+GC, likely bonded nitrate, while the other peak at 1046 $cm^{-1}$ was attributable to unbonded aqueous nitrate (Liang et al., 2022a), which can undergo photolysis to form a wealth of oxidants that lead to glycine decay (Figure 3a).


Furthermore, glycine can be ionized into different forms according to the local conditions, including cationic, neutral, and anionic of different reactivities. Neutral glycine is predominantly in its zwitterionic form in an aqueous solution (Aikens et al., 2006). The rate of anionic glycine reacting with OH radicals is 2-orders of magnitude higher than that of zwitterionic glycine (Berger et al., 1999; Buxton et al., 1988), while that of zwitterionic glycine is several times higher than cationic glycine. These

differences were due to the increased energy barriers for oxidation upon protonation (Wen et al., 2022).

However, the ionization of glycine in different environments is still very controversial, especially under nonideal conditions, despite numerous experimental and computational efforts (Pérez De Tudela et al., 2016; Tortonda et al., 1996). The ionic forms of glycine were found pH-dependent. Cationic, zwitterionic, and anionic dominate the regimes of pH<2.34, 2.34<pH<9.6, and

pH>9.6, respectively. The initial pH of SN+GC particles and AN+GC particles are 6.41±0.34 and 5.95±0.28, respectively. In diluted aqueous solution, glycine was considered zwitterion at these pHs. However, it has been proposed that 5-9 water molecules are required to stabilize a zwitterionic glycine molecule (Pérez De Tudela et al., 2016; Vyas et al., 2011; Xu et al., 2003), which may not be applicable to concentrated particles with fewer water molecules than the diluted solution. The water-to-glycine mole ratios were estimated to be about 6 and 2 in AN+GC and SN+GC particles at 80% RH, respectively. The lower

availability of water molecules in SN+GC particles results in less zwitterionic glycine and more non-ionized glycine than in AN+GC particles. The $-NH_2$ was unprotonated in non-ionized glycine and more susceptible to oxidation, which might promote glycine decay in SN+GC particles. Reduction of glycine reactivity due to the bounding with nitrate was also possible.

### 3.4 The dependence of glycine decay on the initial solute concentration and phase state

As discussed above, the distinct photochemistry between AN+GC and SN+GC particles may likely be related to the solvation and molecular interactions. One would expect that these effects are more evident at lower RH (but before crystallization), with higher solute concentrations and fewer water molecules. Figure 4a shows the percentage GC decay after irradiation as a function of the initial solute concentrations.



At 0.01M, the percentage GC decay is approximately 5% in both AN+GC and SN+GC solutions (Figure 4a). However, as the initial solute concentration increased from 0.01 M to ~7.6 M, the percentage GC decay in SN+GC particles increased by more than one order of magnitude to 70%, while that of the AN+GC particles remained small. The apparent nitrate photolysis rate constant J shows good correlation with the percentage GC decay ($R^2$ = 0.99, Figure 4b), which may suggest that nitrate photolysis is the key driver for the glycine decay, though the increased ionic strength might also play a promotive role

(Herrmann et al., 2015).

   The increased J in SN+GC particles with initial concentration can be potentially explained by the reduced solvent cage effects at high nitrate concentrations (Gen et al., 2022). Specifically, the fragments generated from nitrate photolysis are initially surrounded by a cage of water molecules. Their diffusion out of the cage competes with the regeneration of nitrate anions by

recombining the fragment. Hence, fewer surrounding water molecules in the concentrated particles resulted in reduced recombination and higher J. The absence of an increase in J with increasing initial concentration in AN+GC particles can be explained by the suppressed nitrate photolysis due to molecular interactions, which likely counteracted the promotion by reduced solvent cage etc.

Though no phase transition occurred at RH below 60%, the percentage GC decay in AN+GC particles was very small (Figure 4c). On the other hand, SN+GC particles crystallized at 50%, and the percentage GC decay after 8 h irradiation became <5%, much smaller than that at 80% RH (Figure 4c). Such reduction in glycine decay was likely due to the ineffective photolysis of nitrate in the crystalline lattices (Asher et al., 2011). The crystalline lattices greatly constrain the diffusion of nitrate photolysis products and facilitate their recombination to form nitrate, resulting in a very low photolysis quantum yield.


## 4 Atmospheric implications

   This work showed the distinct decay characteristics of glycine as a model FAA in AN and SN particles under the light. AN+GC particles did not crystalize at RH as low as 3%, while SN+GC did at 50-60% RH. On the other hand, glycine in AN+GC particles exhibited a much slower decay than in SN+GC particles under UV irradiation. A plausible explanation was the water-

mediated bonding between nitrate and GC in AN+GC particles that suppressed crystallization but also hindered nitrate photolysis from generating oxidants and reduced the reactivity of glycine. In contrast, some unbonded nitrate existed in deliquescent SN+GC particles to undergo photolysis and triggered glycine decay effectively, though it was significantly hindered once the particle crystallized. Besides glycine, alanine (Ala) was another major FAA in atmospheric particles (mole ratio of Ala to total FAAs = 0.07-0.17) (Matos et al., 2016; Zhu et al., 2020; Zhang et al., 2003). After 8 h irradiation, we also

found evident Ala decay in deliquescent SN+Ala particles, but not AN+Ala ones (Figure S8).

It is widely reported that nitrate dominantly exists as SN in the coarse mode aged sea salt particles and AN in the fine mode particles (Zhuang et al., 1999a, b). Concentrations of FAAs in the coarse mode were ~10 times higher than that in the fine mode (Helin et al., 2017), which implies that the FAAs plausibly co-existed with SN in atmospheric particles and subjected to oxidation triggered by effective SN photolysis.

Similar to what we found in SN+GC particles, it has been reported that particulate nitrate photolysis rate constants (i.e., $10^{-5}$ $s^{-1}$) can be 2-orders of magnitude higher than nitrate photolysis in cloud and fog water (i.e., $10^{-7}$ $s^{-1}$), likely due to the reduced surface cage, etc. (Gen et al., 2022). $H_2O_2$ photolysis ($J = ~2 \times 10^{-6}$ $s^{-1}$) was considered the primary OH source in aqueous cloud water (Bianco et al., 2015), and the aqueous-phase reactions with OH were reported as an important sink of the cloud FAAs (Wen et al., 2022). However, the small liquid water content in aerosol particles limits the partitioning of $H_2O_2$ into the particle phase. Taking particulate $[H_2O_2]$ and $[NO_3^-]$ as 0.1 ng $\mu g^{-1}$ and 0.2 $\mu g$ $\mu g^{-1}$ (per $PM_{2.5}$ mass) (Xuan et al., 2020; Cheng et al., 2016; Li et al., 2022b), respectively, the OH generation from particulate nitrate photolysis could be 4-orders of magnitude higher than from $H_2O_2$ photolysis. Other oxidants from nitrate photolysis, such as nitrite and HONO, can also react with FAAs to promote their decay, via N-nitration (Kitada et al., 2020).

Overall, our work shed light on the potential role of particulate nitrate photolysis in the sink of the atmospheric FAAs, which impacts the cycling of atmospheric organic nitrogen. Further investigations are encouraged to explore the detailed reaction mechanism and the molecular interaction patterns. The reaction rate constants between FAAs and different oxidants from nitrate photolysis can further help quantify the contribution of nitrate photolysis in FAA degradation and improve the prediction of the atmospheric lifetime of FAAs.

**Data availability**

Supplementary figures were provided.

**Author contributions**

ZL and CKC conceptualized the study, ZL, ZC and RZ performed the experiments and analyzed the data, ZL wrote the manuscript. ZL, ZC, RZ and CKC reviewed and edited the manuscript.

**Competing interests**

The authors declare that they have no conflict of interest.

**Acknowledgment**

We gratefully acknowledge the support from the Hong Kong Research Grants Council (No. 11314222, R1016-20F), and the National Natural Science Foundation of China (No. 42275104, 41905122).



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



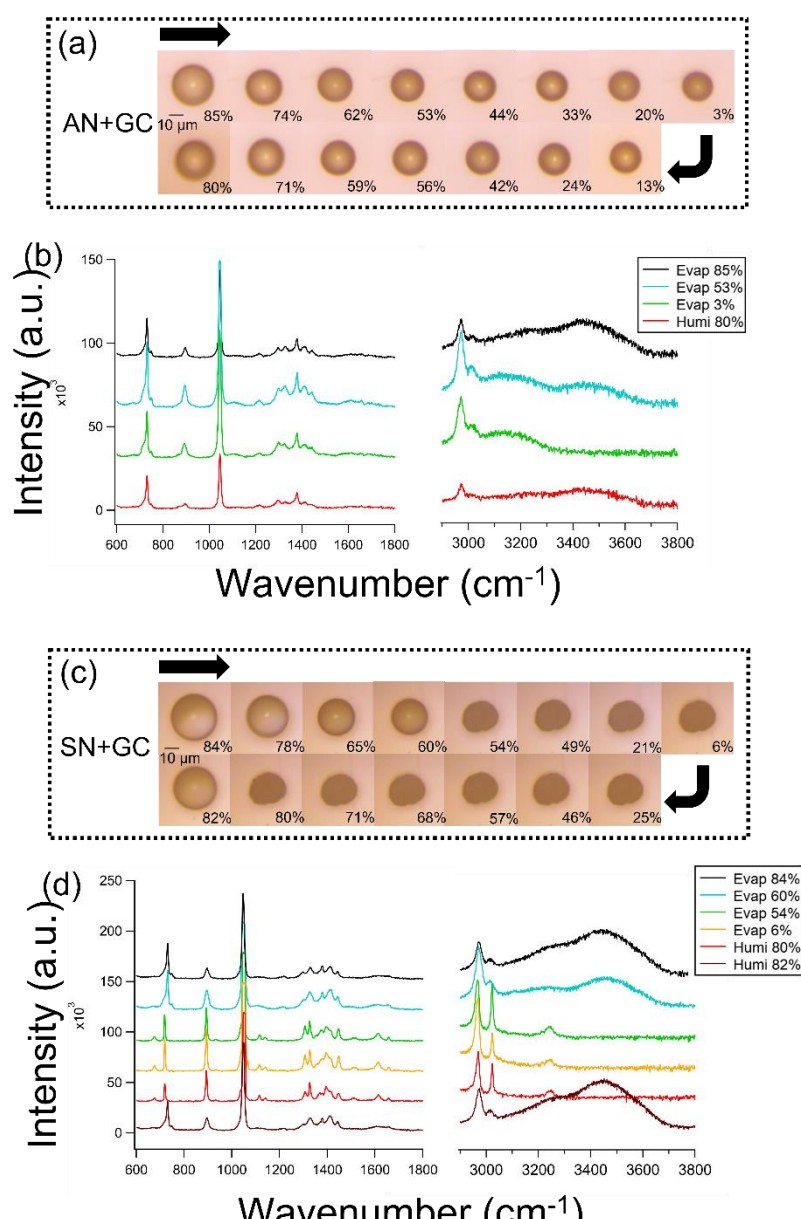


**Figure 1.** Images and Raman spectra of the mixed (a, b) AN+GC particles and (c, d) SN+GC particles during an evaporation-humidification cycle. The arrows in a and c show the changes of relative humidity.



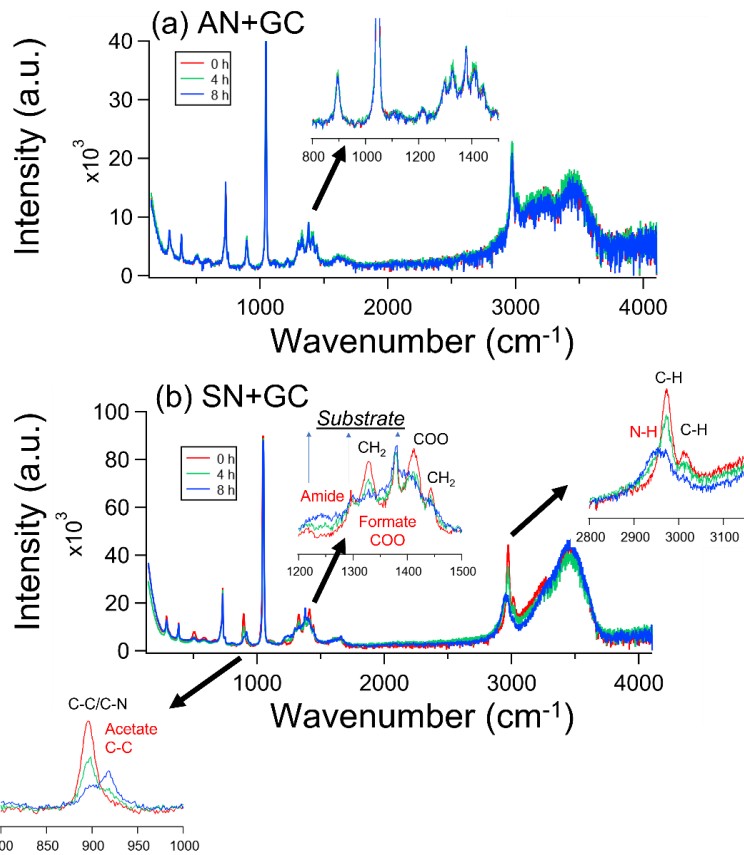


**Figure 2.** Raman spectral evolution of (a) AN+GC and (b) SN+GC particles after 0-, 4-, and 8-hours irradiation. Insets are expanded regions of the SN+GC particle spectra in the ranges of [860, 960], [1150, 1600], and [2800, 3150] (unit: cm$^{-1}$). The red annotations denote the peaks from products. The spectra were normalized by the substrate peak at around 1400 cm$^{-1}$.






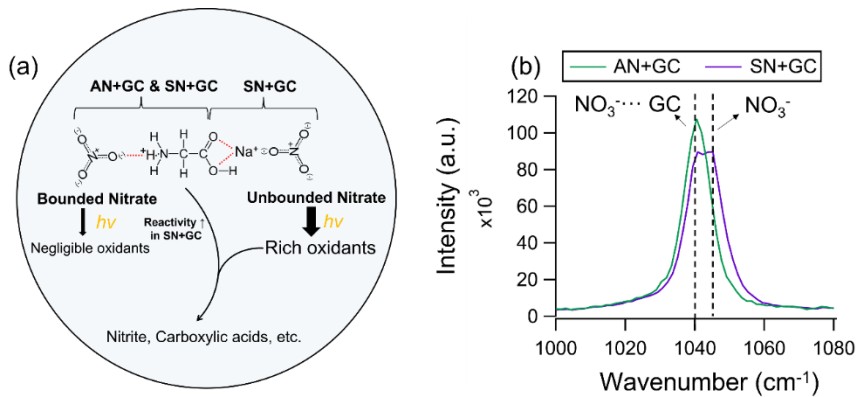

**Figure 3.** (a) A plausible schematic of the photochemical processes and molecular interactions in AN+GC and SN+GC
particles. Red dashed lines indicate the binding. The binding between AN+GC may be mediated by water (Ashraf et al., 2021).
(b) The Raman peak of nitrate of AN+GC and SN+GC particles at 80% RH before illumination. The spectra were normalized
by the substrate peaks.


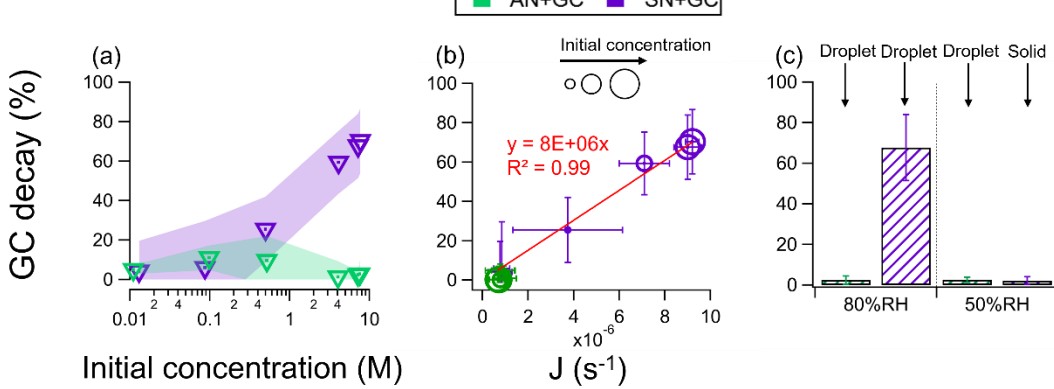

**Figure 4.** (a) The percentage decay of GC as a function of the initial concentration of GC in AN+GC and SN+GC equimolar
mixtures. The shade regions represent the standard deviations. (b) The correlation between nitrate photolysis rate constant (J)
and the percentage GC decay. (c) The percentage GC decay in AN+GC and SN+GC particles at 80% and 50% RH.