# Peer review of "Distinct Photochemistry in Glycine Particles Mixed with Different Atmospheric Nitrate Salts"

_EGUsphere, 2023_

## Referee Comment (RC1)

In this paper, in-situ Raman technique was used to research the photochemistry in nitrate-glycine mixed particles at various RHs. The apparent nitrate photolysis rate constants and percentages GC decay were obtained. The phase transition behaviors of nitrate-glycine were obtained, which showed the role of molecular interaction in determining the physicochemical properties and chemical reactivity of particles. In AN+GC mixed particles, the glycine photochemistry is negligible, and nitrate photochemistry is weak. But in SN+GC mixed particles, products of nitrate and glycine photochemistry, $HNO_2/NO_2^-$, amide, ammonia or/and amine are detected, and the apparent nitrate photolysis rate constant is 4.5-folds higher than that of AN+GC particles.

Questions and comments:

1. In line 57, the word "behaviors" should be deleted.

2. In line 134, the number "1/2" in equation (6) should be removed.

3. In line 136, "Equation 9" does not exist, it should be changed to "Equation 7".

4. In line 139, "/" in equation (7) is misleading, it is better to change it to "or".

5. In this paper, the mole ratio of 1:1 for glycine and nitrate is used in all the experiments, but the photochemistry of pure glycine solution and pure nitrate solution are not studied. If the photochemistry of pure species is missing, how can we conclude that there exists an interaction between glycine and nitrate in the mixed particles affecting their photolysis?

6. In line 232, the viewpoint "The apparent nitrate photolysis rate constant $J$ shows good correlation with the percentage GC decay ($R^2$=0.99, Figure4b), which may suggest that nitrate photolysis is the key driver for the glycine decay" is proposed. But photolysis rate constant $J$ is determined by illumination intensity, and according to Eq.2, $J$ is independent on solute concentration. In Figure 4b, various apparent nitrate photolysis rate constants are displayed and the illuminant of 300nm LED lamp is used in the experiments, so what is the definition of apparent nitrate photolysis rate constant in this paper? Is the x-axis label wrong in Figure 4b? And should it be change to nitrate photolysis rate instead?

---

## Referee Comment (RC5)

Slips of the pen both in sentences and equations have been rectified, and the answers of questions are satisfactory. The manuscript can be considered for publication.

---

## Author Comment (AC1)

**Reply on RC1:**

*In this paper, in-situ Raman technique was used to research the photochemistry in nitrate-glycine mixed particles at various RHs. The apparent nitrate photolysis rate constants and percentages GC decay were obtained. The phase transition behaviors of nitrate-glycine were obtained, which showed the role of molecular interaction in determining the physicochemical properties and chemical reactivity of particles. In AN+GC mixed particles, the glycine photochemistry is negligible, and nitrate photochemistry is weak. But in SN+GC mixed particles, products of nitrate and glycine photochemistry, $HNO_2/NO_2^-$, amide, ammonia or/and amine are detected, and the apparent nitrate photolysis rate constant is 4.5-folds higher than that of AN+GC particles.*

**Authors' Response:** Thank you so much for the constructive comments. Kindly please find our responses below accordingly.

*Questions and comments:*

*In line 57, the word "behaviors" should be deleted.*

**Authors' Response:** The word "behaviors" has been removed.

*In line 134, the number "1/2" in equation (6) should be removed.*

**Authors' Response:** As WGR is the water-to-glycine mole ratio, and the dry solute contains 1:1 (mole ratio) glycine and nitrate, the equation was revised to:

$$WGR = 2 \times \frac{n_w}{n_{dry}} = 2 \times [\frac{(d_{wet})^3}{(d_{dry})^3} \times \frac{\rho_{wet}}{\rho_{dry}} - 1] \times \frac{M_{dry}}{M_w} \qquad (5)$$

*In line 136, "Equation 9" does not exist, it should be changed to "Equation 7".*

**Authors' Response:** Sorry for the confusion. The equation number has been corrected.

*In line 139, "/" in equation (7) is misleading, it is better to change it to "or".*

**Authors' Response:** Agree. We have changed it to "or"

*In this paper, the mole ratio of 1:1 for glycine and nitrate is used in all the experiments, but the photochemistry of pure glycine solution and pure nitrate solution are not studied. If the photochemistry of pure species is missing, how can we conclude that there exists an interaction between glycine and nitrate in the mixed particles affecting their photolysis?*

**Authors' Response:** The absorption of glycine is below 260 nm, and we found no glycine decay in pure GC particles without nitrate under the light. The nitrate photolysis rate constant in SN+GC particles was comparable to pure SN particles without glycine ($1.2\times10^{-5}\,s^{-1}$), which indicates that glycine has a minor suppression effect on SN photolysis. We have clarified this point in the revised manuscript.

> **Line 86-89:** Glycine absorbs light at below 260 nm, but it can form light-absorbing meso-clusters in droplets and trigger photosensitization to degrade themselves at 532 nm, the Raman excitation wavelength in our experiments (Ishizuka et al., 2023). However, this mechanism plays a minor role in our system as no glycine decay in GC droplets without nitrate at 80%RH was found.

> **Line 193-194:** The efficiencies of nitrate photolysis in the two mixed systems were different. The fitted apparent nitrate photolysis rate constant of SN+GC particles at 80% RH was $9 \times 10^{-6}\,s^{-1}$ ($R^2 = 0.95$), 4.5-folds higher than that of AN+GC particles (Figure S7). The nitrate photolysis rate constant in SN+GC particles was comparable to SN particles without glycine ($1.2\times10^{-5}\,s^{-1}$), which indicates that glycine has a minor suppression effect on SN photolysis.

*In line 232, the viewpoint "The apparent nitrate photolysis rate constant J shows good correlation with the percentage GC decay (R2=0.99, Figure4b), which may suggest that nitrate photolysis is the key driver for the glycine decay" is proposed. But photolysis rate constant J is determined by illumination intensity, and according to Eq.2, J is independent on solute concentration. In Figure 4b, various apparent nitrate photolysis rate constants are displayed and the illuminant of 300nm LED lamp is used in the experiments, so what is the definition of apparent nitrate photolysis rate constant in this paper? Is the x-axis label wrong in Figure 4b? And should it be change to nitrate photolysis rate instead?*

**Authors' Response:** As explained in Equation 2, J is the first-order nitrate photolysis rate constant. It reflects the efficiency of photolysis, which is affected by the light intensity, the quantum yield, and the absorption cross-section of the chemical species. The correlation between J and percentage GC decay indicates how the efficiency of nitrate photolysis influences the decay of glycine under different experimental conditions.

> **Line 112-113:** The apparent nitrate photolysis rate constant J ($s^{-1}$) was estimated as Eq. 2. J depends on the light intensity, quantum yield, and absorption cross-section of nitrate (George et al., 2015).

**Reference**

George, C., Ammann, M., D'Anna, B., Donaldson, D. J., and Nizkorodov, S. A.: Heterogeneous photochemistry in the atmosphere, Chem Rev, 115, 4218-4258, 10.1021/cr500648z, 2015.

Ishizuka, S., Reich, O., David, G., and Signorell, R.: Photo-Induced Shrinking of Aqueous Glycine Aerosol Droplets, Atmos. Chem. Phys. Discuss., 2023, 1-16, 10.5194/acp-2023-6, 2023.

---

## Author Comment (AC2)

**Reply on RC2:**

*General:*

*The authors presented experimental results of (1) phase transition behaviors and (2) photochemical degradation of mixed particles with glycine (GC) and ammonium nitrate (AN) or sodium nitrate (SN). Experiments were conducted in a custom-built flow cell reactor with deposited particles, aided by in-situ Raman characterization and off-line chemical analysis. As relative humidity (RH) varied, the mixed particles underwent phase transitions in line with observations in literature. The photochemical behaviors (300-nm light illumination) of AN + GC particles and SN + GC particles, however, are distinctly different, with the latter being much more reactive than the former. The authors proposed that the availability of more abundant "free" nitrate ions in the SN + GC system (where nitrate is not strongly bonded with GC, and binding between carboxylic group of GC and sodium ion "frees up" the nitrate ions) might be the reason behind, as nitrate photolysis can generate an array of oxidants to oxidize GC. This statement was supported by Raman observation that SN + GC particles showed higher signal intensity of "free" nitrate ions. It was implied from this observation that GC, or more generally free amino acids, might decay faster in Na-rich particles (e.g., coarse sea spray aerosols) than in ammonium-rich particles (e.g., urban aerosols). The experiments were well designed and conducted, and arguments in the manuscript are also well articulated. I therefore recommend Minor Revision before publication.*

**Authors' Response:** Thank you so much for your valuable comments. Kindly please find our responses accordingly below.

*Specific:*

*P7, reason(s) for faster photo-degradation of SN + GC particles. The authors presented at least three possible reasons to explain the observed faster decay of GC in the SN + GC particles than in the AN + GC particles: (1) more abundant "free" nitrate ions (the first paragraph in P7), (2) ionic form of GC (anion, zwitterion, or cation) (the second paragraph in P7), and (3) water-to-glycine molar ratio (the third paragraph in P7). It is not clear how these potential reasons are related to each other. That is, are they in parallel, being all possible, with the first one the most important (as the authors stated), and they might even be competing? Or are they intertwining to result in the observed result, i.e., the zwitterion ion (reason 2) form promotes the formation of "free" nitrate ions (reason 1)? Clarification of this might be helpful in understanding what other inorganic cations (e.g., potassium, magnesium etc.) and other free amino acids (what ion form is prevailing in relevant pH range) will behave in similar photochemical processes. In addition, the last sentence in L222 reads ambiguous. Not sure whether it is referring to SN + GC particles or AN + GC particles. If the former, it is contradictory to previous statements; if the latter, please specify.*

**Authors' Response:** Thank you for the insightful comments. The "free nitrate" affords nitrate photolysis to generate oxidants for glycine, while the ionic form of glycine and its availability may modulate the reactivity of glycine. They are all possible consequences of the molecular

configuration in the concentrated AN+GC and SN+GC systems. However, the strong linear relationship between nitrate photolysis rate constant and percentage glycine decay implies that the efficiency of nitrate photolysis is more important than the reactivity of glycine in determining the overall decay rate of glycine. We have expanded our discussion on this point as follows.

> **Line 216-235:** The apparent nitrate photolysis rate constant J shows a good correlation with the percentage GC decay ($R^2 = 0.99$, Figure 4b), which suggests that nitrate photolysis is the key driver for the glycine decay.
>
> The different reactivity of glycine between SN+GC and AN+GC particles may also contribute to the distinct photochemistry. For instance, glycine can be ionized into different forms according to the local conditions, including cationic, zwitterionic, and anionic of different reactivities (Aikens et al., 2006). Zwitterionic denotes the charge-separated form amino acids in aqueous solutions and in crystalline states (e.g., $NH_3^+$–$CH_2$–$COO^-$). Several possible zwitterionic conformers of glycine have been proposed with the addition of 1-3 water molecules (Krauklis et al., 2020). The rate of anionic glycine reacting with OH radicals is 2-orders of magnitude higher than that of zwitterionic glycine (Berger et al., 1999; Buxton et al., 1988), while that of zwitterionic glycine is several times higher than cationic glycine. These differences were due to the increased energy barriers for oxidation upon protonation (Wen et al., 2022). However, the protonation constants of glycine in concentrated solution were difficult to define. Qualitatively, a possible lower degree of glycine protonation in SN+GC particles than AN+GC particles might enhance the reactivity of glycine.
>
> We also note that the initial water-to-glycine mole ratios were higher for AN+GC particles (6) than SN+GC particles (2), and sodium has higher hydration number (6)(Medoš et al., 2019) than ammonium (4)(Guo et al., 2020). Therefore, the availability of free water in AN+GC particles is likely higher than in SN+GC particles. This could affect the configuration of glycine dimers or trimers, such as the possible complexation or charge interactions between the anionic carboxylate and the cationic -$NH_3^+$ groups. These factors could also modulate the photo-reactivity of glycine.

The pHs of the mixed GC nitrate particles at 80% RH were around 6 according to the pH paper (Craig et al., 2018), at which zwitterionic glycine is the most abundant form. Without any information of possible unknown molecular interactions that affect the protonation equilibrium, we assume that glycine is mostly in the zwitterionic form. As shown in Figure 3a, the complexation of glycine with the cation is crucial in allowing free nitrate for photolysis. Other atmospheric cations, such as potassium, magnesium and calcium, can also form complexes with the carboxylic group of amino acids (Case et al., 2020; Lester et al., 2010; Tang et al., 2016). The pH of global ambient aerosol can span from 0-6 (Weber et al., 2016; Liu et al., 2017), which means that amino acids can exist in both cationic (0<pH<2) and zwitterionic (2<pH<10) forms (Locke et al., 1983; Stroud et al., 1983). Previous studies have reported that cationic amino acids have enhanced complexation with metal cations due to the protonation (Moision et al., 2002). We have incorporated this information into the revised manuscript.

**Line 266-271:** As shown in Figure 3a, the complexation of glycine with the cation is crucial for allowing free nitrate for photolysis. Other atmospheric cations, such as potassium, magnesium and calcium, can also form complexes with the carboxylic group of amino acids (Case et al., 2020; Lester et al., 2010; Tang et al., 2016). The pH of global ambient aerosol can span from 0-6 (Weber et al., 2016; Liu et al., 2017), which means that amino acids can exist in both cationic ($0<pH<2$) and zwitterionic ($2<pH<10$) forms (Locke et al., 1983; Stroud et al., 1983). Previous studies have reported that complexation is enhanced for cationic amino acids due to the protonation (Moision et al., 2002).

*P9, implications. What causes the stronger binding between sodium ion with the carboxylic group compared to that between ammonium ion with the carboxylic group. Is it because sodium is a stronger alkaline species than ammonium? Or due to some sort of indirect effect (e.g., how much solvated the carboxylic group is) from the low water-to-glycine ratio in the SN + GC system? In ambient aerosol particles, there might be other alkaline species too (potassium, magnesium, amines etc.). It would be good to comment on whether alkalinity of degree of solvation might lead to such increased photodegradation of free amino acids, if possible.*

**Authors' Response:** Unlike metal cations, ammonium lacks vacant orbitals to coordinate with the amino acid functional groups. Hence ammonium cannot form stable complexes with amino acids, but only protonate them. We agree that the effect of alkalinity and solvation level on the fate of amino acids upon nitrate photolysis is an interesting topic for future research, and we have mentioned it in the revised manuscript.

**Line 273-280:** Overall, our work sheds light on the potential role of particulate nitrate photolysis in the sink of the atmospheric FAAs, which impacts the cycling of atmospheric organic nitrogen. The reaction rate constants between FAAs and different oxidants from nitrate photolysis can further help quantify the contribution of nitrate photolysis in FAA degradation and improve the prediction of the atmospheric lifetime of FAAs. The reactivity analysis in concentrated systems is complex, and our experimental results can provide valuable data to parameterize the complex thermodynamics in future studies. Systematic studies of the detailed molecular mechanism and the factors influencing nitrate photochemistry and FAA decay, such as molecular configuration, alkalinity, and solvation are recommended. Quantum chemical and molecular dynamic simulations with appropriate parameters would be useful tools for this purpose.

*Technical:*

*P1/L13: add "of" before "glycine".*

**Authors' Response:** We have added an "of" before "glycine".

*P2/L49: remove "(Wen et al., 2022)" at the end of the sentence.*

**Authors' Response:** We have removed "(Wen et al., 2022)" from the end of the sentence.

*P2/L52: remove "behavior.".*

**Authors' Response:** We have removed "behaviors".

*P4/L109: revise citation format of "Matsumoto et al….".*

**Authors' Response:** We have revised the format of "Matsumoto et al.".

*P5/L135: I do not see Equation 9.*

**Authors' Response:** Sorry for the confusion, we have corrected the equation number.

*P6/L171: should NH3 be -NH2 or -NH3+? Are you referring to the amino group of GC?*

**Authors' Response:** We have revised the $NH_3$ to $-NH_3^+$.

**Reference**

Aikens, C. M. and Gordon, M. S.: Incremental Solvation of Nonionized and Zwitterionic Glycine, Journal of the American Chemical Society, 128, 12835-12850, 10.1021/ja062842p, 2006.

Berger, P., Karpel Vel Leitner, N., Doré, M., and Legube, B.: Ozone and hydroxyl radicals induced oxidation of glycine, Water Research, 33, 433-441, https://doi.org/10.1016/S0043-1354(98)00230-9, 1999.

Buxton, G. V., Greenstock, C. L., Helman, W. P., and Ross, A. B.: Critical Review of rate constants for reactions of hydrated electrons, hydrogen atoms and hydroxyl radicals (·OH/·O− in Aqueous Solution, Journal of Physical and Chemical Reference Data, 17, 513-886, 10.1063/1.555805, 1988.

Case, D. R., Zubieta, J., and P. Doyle, R.: The Coordination Chemistry of Bio-Relevant Ligands and Their Magnesium Complexes, Molecules, 25, 3172, 2020.

Craig, R. L., Peterson, P. K., Nandy, L., Lei, Z., Hossain, M. A., Camarena, S., Dodson, R. A., Cook, R. D., Dutcher, C. S., and Ault, A. P.: Direct determination of aerosol pH: Size-resolved measurements of submicrometer and supermicrometer aqueous particles, Analytical chemistry, 90, 11232-11239, 2018.

Guo, J., Zhou, L., Zen, A., Michaelides, A., Wu, X., Wang, E., Xu, L., and Chen, J.: Hydration of ${\mathrm{NH}}_{4}^{+}$ in Water: Bifurcated Hydrogen Bonding Structures and Fast Rotational Dynamics, Physical Review Letters, 125, 106001, 10.1103/PhysRevLett.125.106001, 2020.

Krauklis, I. V., Tulub, A. V., Golovin, A. V., and Chelibanov, V. P.: Raman Spectra of Glycine and Their Modeling in Terms of the Discrete–Continuum Model of Their Water Solvation Shell, Optics and Spectroscopy, 128, 1598-1601, 10.1134/S0030400X20100161, 2020.

Lester, G. E., Jifon, J. L., and Makus, D. J.: Impact of potassium nutrition on postharvest fruit quality: Melon (Cucumis melo L) case study, Plant and soil, 335, 117-131, 2010.

Liu, M., Song, Y., Zhou, T., Xu, Z., Yan, C., Zheng, M., Wu, Z., Hu, M., Wu, Y., and Zhu, T.: Fine particle pH during severe haze episodes in northern China, Geophysical Research Letters, 44, 5213-5221, 2017.

Locke, M. J. and McIver Jr, R. T.: Effect of solvation on the acid/base properties of glycine, Journal of the American Chemical Society, 105, 4226-4232, 1983.

Medoš, Ž., Plechkova, N. V., Friesen, S., Buchner, R., and Bešter-Rogač, M.: Insight into the Hydration of Cationic Surfactants: A Thermodynamic and Dielectric Study of Functionalized Quaternary Ammonium Chlorides, Langmuir, 35, 3759-3772, 10.1021/acs.langmuir.8b03993, 2019.

Moision, R. M. and Armentrout, P. B.: Experimental and Theoretical Dissection of Sodium Cation/Glycine Interactions, The Journal of Physical Chemistry A, 106, 10350-10362, 10.1021/jp0216373, 2002.

Stroud, E. D., Fife, D. J., and Smith, G. G.: A method for the determination of the pKa of the. alpha.-hydrogen in amino acids using racemization and exchange studies, The Journal of Organic Chemistry, 48, 5368-5369, 1983.

Tang, N. and Skibsted, L. H.: Calcium Binding to Amino Acids and Small Glycine Peptides in Aqueous Solution: Toward Peptide Design for Better Calcium Bioavailability, Journal of Agricultural and Food Chemistry, 64, 4376-4389, 10.1021/acs.jafc.6b01534, 2016.

Weber, R. J., Guo, H., Russell, A. G., and Nenes, A.: High aerosol acidity despite declining atmospheric sulfate concentrations over the past 15 years, Nature Geoscience, 9, 282-285, 2016.

Wen, L., Schaefer, T., Zhang, Y., He, L., Ventura, O. N., and Herrmann, H.: T- and pH-dependent OH radical reaction kinetics with glycine, alanine, serine, and threonine in the aqueous phase, Physical Chemistry Chemical Physics, 24, 11054-11065, 10.1039/D1CP05186E, 2022.

---

## Author Comment (AC3)

**Reply on RC3:**

*General comments:*

*The paper is concerned with a problem which is certainly of environmental interest, but which nevertheless touches problems in concentrated solutions, which are beyond the reach of classical thermodynamic models. Under conditions of such low solvent availability, the activity coefficients can explode to values >1000, making the reactivity analysis very complex. I am personally in favor of making experiments other than the classic ones, such as in this paper. However, precisely because of this, interpretation can be difficult.*

**Authors' Response:** Thank you so much for your constructive comments. The reactivity analysis in concentrated systems is complex, and our experimental results can provide valuable data to parameterize the complex thermodynamics in future studies. We also agreed with the reviewer that the interpretation of our findings should be further improved. Kindly please find our responses to the comments accordingly below:

*Amino acids in real conditions (in aqueous solutions and in crystalline form) are separate charge systems (NH+3–CH2–COO–). Several possible zwitterionic conformers of glycine have been calculated with the addition of 1-3 water molecules [Opt. Spectrosc. 128, 1598–1601 (2020). https://doi.org/10.1134/S0030400X20100161]. In both low water content and crystalline form GLY is ZW (NH+3–CH2–COO–), as in water, always (and pH only changes the fractional amount of charges from positive to negative). In several parts of the article GLY is referred to as neutral/non-ionized/less zwitterionic. This is a confusion, because the form NH2–CH2–COOH does not exist. Thus, the paper's key conclusion given on line 220 is inconsistent. Also, on the same lines, it is stated that "The -NH2 was unprotonated…and more susceptible to oxidation". This is only possible at pH above 10, so the observed reactivity must have other explanations. In such a concentrated solution the pH is also difficult to define, and no idea of the actual protonation constants is reported. Thus, if it is true that the reactivity of OH is greater for anionic GLY, the "formal" pH would be basic. Are there any hypotheses about it?*

**Authors' Response:** Aerosol pH was defined as the activity of hydrogen ions ($H^+$) in aqueous aerosol particles (Buck et al., 2002; Li et al., 2022; Peng et al., 2019). The pHs of the mixed GC nitrate particles at 80% RH were around 6 according to the pH paper measurements (Craig et al., 2018), at which zwitterionic glycine is the most abundant form. Without any information of possible unknown molecular interactions that affect the protonation equilibrium (i.e., protonation constants), we agree with the reviewer that glycine is mostly in the zwitterionic form and have removed the terminology of neutral and non-ionized to avoid confusion. We also added further information on zwitterion as the reviewer suggested.

> **Line 221-224:** For instance, glycine can be ionized into different forms according to the local conditions, including cationic, zwitterionic, and anionic of different reactivities (Aikens et al., 2006). Zwitterionic denotes the charge-separated form amino acids in aqueous solutions and in crystalline states (e.g., $NH_3^+–CH_2–COO^-$). Several possible

zwitterionic conformers of glycine have been proposed with the addition of 1-3 water molecules (Krauklis et al., 2020).

As shown in Figure 4b, the strong linear correlation between nitrate photolysis rate constant and percentage glycine decay suggests that the efficiency of nitrate photolysis is the main factor for the glycine decay, which is modulated by the interaction between nitrate and glycine. We mentioned the protonation and configuration of glycine as a possible minor factor that could affect the photochemistry, while also acknowledging the difficulty of determining the protonation constants of glycine in concentrated solutions, as pointed out by the reviewer. Therefore, we have added this information to the revised manuscript with more explanation.

> **Line 216-229:** The apparent nitrate photolysis rate constant J shows good correlation with the percentage GC decay ($R^2 = 0.99$, Figure 4b), which suggests that nitrate photolysis is the key driver for the glycine decay.
>
> The different reactivity of glycine between SN+GC and AN+GC particles may also minorly contribute to the distinct photochemistry. For instance, glycine can be ionized into different forms according to the local conditions, including cationic, zwitterionic, and anionic of different reactivities (Aikens et al., 2006). Zwitterionic denotes the charge-separated form amino acids in aqueous solutions and in crystalline states (e.g., $NH_3^+–CH_2–COO^-$). Several possible zwitterionic conformers of glycine have been proposed with the addition of 1-3 water molecules (Krauklis et al., 2020). The rate of anionic glycine reacting with OH radicals is 2-orders of magnitude higher than that of zwitterionic glycine (Berger et al., 1999; Buxton et al., 1988), while that of zwitterionic glycine is several times higher than cationic glycine. These differences were due to the increased energy barriers for oxidation upon protonation (Wen et al., 2022). However, the protonation constants of glycine in concentrated solution were difficult to define. Qualitatively, a possible lower degree of glycine protonation in SN+GC particles than AN+GC particles might enhance the reactivity of glycine.

*From the GLY ZW nature it follows that the addition of SN or AN could change:*

*1) The availability of free water, as part of it can be solvated by sodium or ammonium cations. This was deduced at line 219, as water to GLY ratio of 6 for AN and 2 for SN. Although it is reasonable as sodium is solvated more than ammonium, these ratios must be further supported by the authors.*

*2) the configuration of the GLY dimer or trimer as some possible complexation or charge interaction is possible both with the anion carboxylate, and the cationic protonated -NH3+ group. This last would change the photoreactivity due to the nitrate, the only absorbing species in the system (the absorption of GLY is below 260 nm, and not involved in the experiments). In Figure 3a the complexation of sodium is depicted, leaving free nitrate supposed to form reactive species. But also a weaker bond with ammonium would lead to the same configuration of free nitrate. Then, what is the ultimate explanation? Overall the paper need a strong revision.*

**Authors' Response:** We observed that the nitrate photolysis rate constant was linearly correlated with the percentage glycine decay, indicating that the efficiency of nitrate photolysis was a key factor for glycine degradation. Moreover, the nitrate photolysis rate constant was significantly larger in SN+GC particles than in AN+GC particles, attributable to the different molecular interactions. Based on the literature, we illustrated the possible differences of interactions of nitrate salts and glycine between AN+GC and SN+GC particles. If the bonds between ammonium or GC and nitrate in AN+GC particles are weaker than those between $Na^+$ or GC and nitrate in SN+GC particles, there will be more free nitrates in AN+GC particles for photolysis than SN+GC particles. However, our Raman data show only one single symmetric nitrate peak in AN+GC particles, which was likely from inactive bonded nitrate, given the small nitrate photolysis rate constant and minor glycine decay. Therefore, the bonds between ammonium or GC with nitrate in AN+GC particulate matrix were likely stronger than that of sodium or GC with nitrate in SN+GC particles. This may explain the suppressed nitrate photolysis in AN+GC particles, compared with SN+GC particles.

The interaction-dependent nitrate photolysis was likely the key driver for glycine decay, but the exact molecular configurations of such interactions still require further work. We also agree with the reviewer that the availability of free water and the configuration of the glycine dimer or trimer may change the photo-reactivity of glycine under nitrate photochemistry. However, we hesitate to speculate too much on the exact molecular configuration since the detailed investigations of quantum chemistry and molecular dynamic simulation with appropriate parameterization for non-ideal solutions is beyond the focus of this paper. We view that such work is better suited for a physical chemistry journal than ACP. We hope that our experimental observations can stimulate researchers to look into the complex interactions described above and outlined by the reviewer.

> **Line 197-235:** For instance, amino acid nitrate can form hydrogen bonding between nitrate from AN and the protonated amino group of glycine (Figure 3a) (Wang et al., 2022; Ashraf et al., 2021). As a result, the amino acids and nitrate ions in the droplet are bounded in an extensive three-dimensional hydrogen-bonded matrix (Wang et al., 2022), in which nitrate photolysis could be hindered (Vimalan et al., 2010). On the other hand, the $COO^-$ of glycine can bind with SN via $Na^+$ directly to form a bidentate complex (Figure 3a) (Moision et al., 2002; Aziz et al., 2008; Selvarani et al., 2022), leaving nitrate unbonded. Nitrate in SN+GC particles has two Raman peaks (Figure 3b). One had the same Raman shift as nitrate in AN+GC, likely bonded nitrate, while the other peak at 1046 $cm^{-1}$ was attributable to unbonded aqueous nitrate (Liang et al., 2022), which can undergo photolysis to form a wealth of oxidants that lead to glycine decay (Figure 3a). The single symmetric nitrate peak, small J, and minor glycine decay of AN+GC particles suggested a negligible fraction of unbonded nitrate. However, we also note that the exact molecular configuration in concentrated particles can be much more complicated than the illustrative example shown in Figure 3a. Detailed investigations of quantum chemistry and molecular dynamic simulation with appropriate parameterization for non-ideal solutions are required.

> One would expect that the molecular interactions are more evident in droplets at lower RH, with higher solute concentrations and fewer water molecules. Figure 4a shows the percentage GC decay after irradiation as a function of the initial solute concentrations. At

[revised manuscript text omitted]